# Estimation of Energy Harvesting by Thermoelectric Cement Composites with Nanostructured Graphene and Metallic Oxides

**Sampad Ghosh [1] and Bidyut Baran Saha [2,3,\*]**

[1]  Department of Electrical and Electronic Engineering, Chittagong University of Engineering and Technology (CUET), Chattogram 4349, Bangladesh; sampad@cuet.ac.bd
[2]  Department of Mechanical Engineering, Kyushu University, 744 Motooka, Nishi-ku, Fukuoka 819-0395, Japan
[3]  International Institute for Carbon-Neutral Energy Research (WPI-I2CNER), Kyushu University, 744 Motooka, Nishi-ku, Fukuoka 819-0395, Japan
\*  Correspondence: saha.baran.bidyut.213@m.kyushu-u.ac.jp; Tel.: +81-92-802-6722

**Abstract:** The measurement of electrical power and efficiency of a thermoelectric generator (TEG) holds significant importance in the realm of thermoelectric materials research and development. The present investigation involves the measurement of thermoelectric characteristics, namely electrical conductivity, Seebeck coefficient, and thermal conductivity, of cement composites containing graphene nanoplatelets and metallic oxides ($Fe_2O_3$, $ZnO$, $MnO_2$). These properties are then utilized to determine the electrical power output and efficiency of the aforementioned composites. It is possible to estimate a power output of up to 1.5 W per square meter when utilizing GnP-ZnO-added cement composites, given a temperature differential of approximately 50 °C. This paper additionally discusses the methodology for fabricating a cement composite-based structural TEG module with the aim of augmenting the overall output voltage, power, and efficiency of the system.

**Keywords:** cement composites; energy harvesting; graphene nanoplatelets; metallic oxides; thermoelectric properties

## 1. Introduction

The world possesses a significant amount of thermal energy that is readily available without any cost. Waste heat, as an example of thermal energy, is generated by sources such as engines, furnaces, boilers, and other similar systems. The sun is a natural source of thermal energy, which is transferred to the earth through the process of heating. The thermoelectric (TE) effect is an efficient way to turn heat energy into electricity. The utilization of structural materials, such as cement, for the purpose of energy harvesting is a highly appealing prospect. People are paying more attention to TE technologies in the area of saving energy in buildings [1,2]. However, due to the inferiority of the conversion efficiency, i.e., the figure of merit, in-building energy harvesting is hard to realize in practice.

For a thermoelectric (TE) material to be effective for energy conversion, it is imperative that it possesses a high figure of merit (ZT). This necessitates a notable Seebeck coefficient, a significant level of electrical conductivity, and a relatively low thermal conductivity. The utilization of carbon and its allotropes is prevalent in cement-based composites due to the significant role of phonons in facilitating heat conduction in carbon, as well as the high degree of phonon scattering observed in the cement matrix. As indicated by reference [3], the incorporation of carbon in cement composites results in a decrease in thermal conductivity. However, a substantial endeavor is required to enhance the Seebeck coefficient and electrical conductivity as opposed to reducing the thermal conductivity. Nanostructured materials are promising to achieve a high-power factor and maintain low heat conductivity, which would improve ZT [4]. Prior research [5–9] used different methods

to improve the ZT of thermoelectric cement composite, with a focus on either increasing the Seebeck coefficient or the electrical conductivity. The ways include making TE cement composites with expanded graphite [5], carbon nanotubes [6,7], graphene nanoplatelets [8], and carbon fibers [9]. Among them, the fascinating attributes of graphene nanoplatelets (GnP) have aided in displaying an improved Seebeck coefficient and electrical conductivity in the cement composite [8]. However, the Seebeck coefficients that were found for GnP-based cement composites were far below 100 $\mu VK^{-1}$.

On the other hand, different kinds of nanoparticles based metallic oxides such as Iron (III) oxide ($Fe_2O_3$), Zinc oxide (ZnO), and Manganese (IV) oxide ($MnO_2$) [10,11] have exhibited a high value of Seebeck coefficient. Nevertheless, it is important to note that the enhancement of thermoelectric properties cannot be solely attributed to metallic oxides due to their restricted electrical conductivity. Graphene nanoplatelets have been integrated with metallic oxides ($Fe_2O_3$, ZnO, and $MnO_2$), resulting in significant improvements in thermoelectric properties, as reported in previous studies [12,13]. However, there were very few works to estimate the harvested electrical energy for the realization of their potential applications.

Thermoelectric (TE) energy harvesting refers to the generation of power, typically at a small scale, through the utilization of external sources of energy that are produced by the sun, waste products, the surrounding environment, or even by the human body. The primary objectives of TE energy harvesting devices with a small-scale design are to enhance convenience and simplicity, reduce costs and increase longevity, ensure dependability, and promote ecological sustainability. Moreover, the utilization of ambient thermal energies presents a promising approach to mitigate the urban heat island (UHI) phenomenon. This means that they are good for the environment and can be used again and again. Small thermoelectric generators have the capability to extract energy from heat sources and can be implemented in various settings, such as a wristwatch [14], a wood stove [15], or a biomass cook stove [16]. Hence, the investigation of thermoelectric energy harvesting devices is highly valuable as it facilitates the provision of power to compact, cordless, and portable electronic devices. In addition, these energy harvesting devices are utilized in autonomous systems and wireless sensor networks [17,18] within intelligent structures, as they possess the ability to operate continuously and autonomously without the need for an external power supply.

In our previous works [12,13], we have observed a significant improvement in the thermoelectric properties of the cement composites with the addition of graphene nanoplatelets and metallic oxides. However, nothing about the estimation of electrical energy and efficiency and possible applications as thermoelectric generators were considered. This study involves the integration of $Fe_2O_3$, ZnO, and $MnO_2$ nanostructured materials into cement composites based on graphene nanoplatelets, followed by an assessment of their thermoelectric characteristics. The obtained thermoelectric properties are then employed to estimate the highest attainable electrical power and efficiency for GnP-oxides-based cement composites as an extension of prior research efforts [12,13]. A maximum electrical power output of approximately 1.5 $Wm^{-2}$ can be obtained through the incorporation of a small quantity of Zinc Oxide and Graphene nanoplatelets (10 weight percent of both ZnO and GnP) into cement composites. The present study also opens a new dimension to enable the realization of a thermoelectric generator (TEG) comprising a p-type and n-type cement composite. The maximum power output of the TEG is estimated to be 0.55 mW when subjected to a temperature differential of 50 °C.

## 2. Materials and Methods

### 2.1. Sample Preparation

The materials utilized in this study were employed in their as-received state without undergoing any additional purification procedures. The procurement of cement and bulk graphene nanoplatelets of H-grade (average particle size: 25 μm, specific density: 2.2 $gcm^{-3}$, thickness: 15 nm, surface area: 50–80 $m^2g$) was carried out from Toyo Matelan

Co. Ltd., Kasugai, Aichi, Japan, and XG Sciences, East Lansing, MI, USA, correspondingly. Metallic oxides, such as Iron (III) oxide ($Fe_2O_3$) (molecular weight: 159.69 gmol$^{-1}$, specific density: 5.15 gcm$^{-3}$, purity: $\geq$99.9%), Zinc oxide (ZnO) (average particle size: $\leq$0.05 µm, molecular weight: 81.39 gmol$^{-1}$, purity: >97%), and Manganese (IV) oxide ($MnO_2$) (average particle size: $\leq$10.0 µm, molecular weight: 86.94 gmol$^{-1}$, specific density: 5.21 gcm$^{-3}$, purity: ~85%), are acquired from Sigma-Aldrich Japan G.K., Higashishinagawa, Shinagawa, Tokyo, Japan.

Cement composite samples were prepared using only GnP, and metallic oxides ($Fe_2O_3$/ZnO/$MnO_2$) and GnP, with a mass ratio of 10 wt% each, were determined to yield optimal results according to [8]. Four samples were prepared with the amount of 10.0 wt% by mass of cement with inclusions of only GnP (10% GnP), and GnP and $Fe_2O_3$ (10% GnP, 10% $Fe_2O_3$), GnP and ZnO (10% GnP, 10% ZnO), and GnP and $MnO_2$ (10% GnP, 10% $MnO_2$), respectively. The dimension of the samples is $2 \times 2 \times 10$ mm$^3$. The methodology for preparing the samples has been explicated in the preceding study [8] and is visually depicted in Figure 1. Initially, the GnP and cement particles were combined using a planetary ball mill (manufactured by Fritsch Japan Co. Ltd., Sumida-ku, Tokyo, Japan) in the presence of 30 g Zirconia balls to ensure uniform dispersion of GnP within the cement. The milling procedure entails a rotational speed of 600 revolutions per minute, a duration of 60 min, a pause of 5 min, and a repetition of 12 cycles. The study utilized a Nitto ANF-30 vibrating sieve machine with a sieve aperture size of 106 µm or 140 mesh to ensure consistent particle sizes and reliable data. The study made sure that the amount of water to cement was always 0.1, irrespective of the concentration of GnP. Subsequently, a combination of various unprocessed constituents was introduced into a cylindrical steel mold, which was then subjected to a compressive force of 40 MPa via a pressing apparatus (NT-200H, NPa system) to produce a compacted product ($\varphi 20 \times 4$ mm$^2$). The resulting material was then allowed to cure under normal atmospheric conditions. Finally, the composite underwent a drying process at a temperature of 200 °C for a duration of 5 h under vacuum conditions, followed by cooling to room temperature to eliminate any residual moisture content. The same procedures are followed to prepare samples for GnP-$Fe_2O_3$, GnP-ZnO, and GnP-$MnO_2$ composites. The cylindrical specimen has been retained for thermal evaluation, whereas the rectangular bar has been extracted from the identical cylindrical specimen to be employed in the four-probe setup. The measurement of the Seebeck coefficient and electrical conductivity is conducted concurrently through the utilization of a four-probe system (RZ2001i) manufactured by Ozawa Science, Naka-ku, Nagoya, Japan. The temperature range for the measurement is set between 25 °C and 75 °C. The study of thermal conductivity was conducted through the utilization of the laser flash technique. This involved the measurement of specific heat capacity and thermal diffusivity using a differential scanning calorimeter (model: DSC-60A, manufacturer: Shimadzu Corporation, Nakagyo-ku, Kyoto, Japan) and a laser flash apparatus (model: LFA 457 MicroFlash, manufacturer: NETZSCH-Gerätebau GmbH, Wittelsbacherstraße, Selb, Germany), correspondingly.

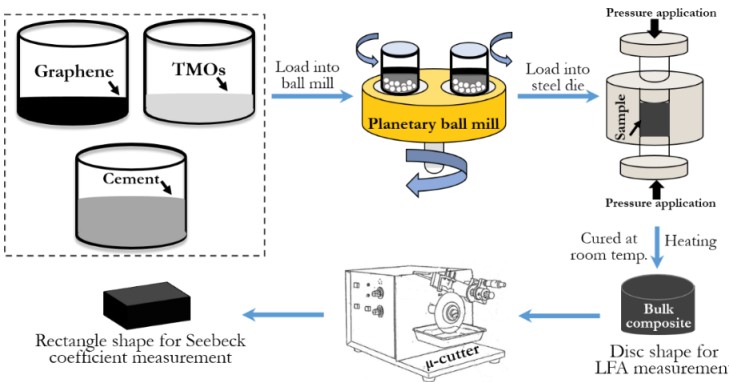

**Figure 1.** Cement composite preparation procedures.

### 2.2. Thermodynamics of a Thermoelectric Generator

The schematic diagram of a thermoelectric generator is depicted in Figure 2, wherein the symbols R, S, and K denote the resistivity, Seebeck coefficient, and thermal conductance of the device, respectively. For the sake of simplicity, it is assumed that the generator's settings will not change with temperature.

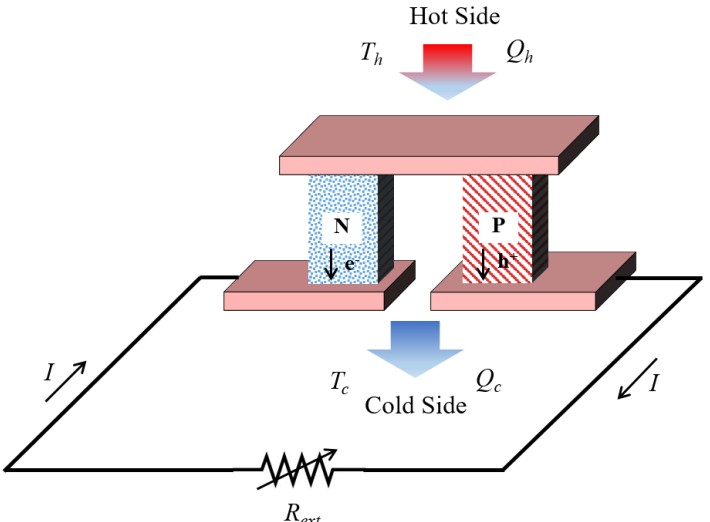

**Figure 2.** Schematic representation of a thermoelectric generator.

The absorbed heat $Q_h$ on the hot side can be mathematically expressed as follows:

$$Q_h = ST_h I - \frac{RI^2}{2} + K\Delta T \tag{1}$$

The second term in the equation represents the Joule heating occurring within the sample, with the assumption that an equal amount of heat is distributed to both sides. The third term accounts for the thermal current's backflow. Likewise, the dissipated heat $Q_c$ on the lower temperature side can be represented as such.

$$Q_c = ST_c I + \frac{RI^2}{2} + K\Delta T \tag{2}$$

The total power output of the thermoelectric module can be expressed in terms of its internal properties, as per the First Law of Thermodynamics.

$$W = Q_h - Q_c = SI(T_h - T_c) - RI^2 \tag{3}$$

An external load resistance can be used to describe the total power flow shown in Figure 2.

$$W = I^2 R_{ext} \tag{4}$$

The total voltage can be obtained by equating Equations (3) and (4).

$$V = IR_{ext} = S\Delta T - RI \tag{5}$$

The electrical current for the module can be derived from Equation (5).

$$I = \frac{S\Delta T}{R + R_{ext}} \tag{6}$$

The magnitude of the electric current (I) remains constant regardless of the quantity of thermocouples present in the system. Upon substitution of the variable "I" into Equation (5), the resulting expression yields the voltage that exists across the module.

$$V = IR_{ext} = \frac{S\Delta T}{R + R_{ext}} R_{ext} \tag{7}$$

The output power, denoted as P, is equivalent to

$$P = VI = \frac{S\Delta T R_{ext}}{R + R_{ext}} \frac{S\Delta T}{R + R_{ext}} = \frac{R_{ext}(S\Delta T)^2}{(R + R_{ext})^2} \tag{8}$$

The maximum electrical power that can be transferred occurs when the external resistance ($R_{ext}$) is equal to the internal resistance (R), resulting in the maximum electrical power output.

$$P_{max} = \frac{(S\Delta T)^2}{4R} = \frac{S^2(T_h - T_c)^2}{4(L/\sigma A)} \tag{9}$$

The sample's length, area, and electrical conductivity are denoted by the variables L, A, and $\sigma$, respectively. The efficiency of thermoelectric conversion can be mathematically represented as the ratio of the power output to the amount of heat taken in at the hot side.

$$\eta = \frac{P}{Q_h} = \frac{\left(1 - \frac{T_c}{T_h}\right)\frac{R_{ext}}{R}}{\left(1 - \frac{R_{ext}}{R}\right) - \frac{1}{2}\left(1 - \frac{T_c}{T_h}\right) + \frac{1}{2ZT}\left(1 - \frac{R_{ext}}{R}\right)^2 \left(1 + \frac{T_c}{T_h}\right)} \tag{10}$$

To achieve the highest possible thermoelectric conversion efficiency, one can derive the conversion efficiency equation in Equation (10) with respect to $R_{ext}/R$ and equate it to zero. Subsequently, the highest attainable conversion efficiency, denoted as $\eta_{max}$, is determined.

$$\eta_{max} = \left(1 - \frac{T_c}{T_h}\right) \frac{\sqrt{(1 + ZT)} - 1}{\sqrt{(1 + ZT)} + \frac{T_c}{T_h}} \tag{11}$$

Therefore, the thermoelectric device relies on two fundamental parameters, namely $P_{max}$ and $\eta_{max}$.

### 3. Results and Discussion

#### 3.1. Thermoelectric Properties

The Seebeck coefficient was not measurable in cement due to its restricted electrical conduction, resulting in both values being regarded as zero. The cement mixture that has graphene nanoplatelets (GnP) in it has an electrical conductivity ($\sigma$) of more than 11 Scm$^{-1}$. Since the conductivity of cement is almost zero, it is anticipated that the charge carriers are contributed predominantly by the graphene nanoplatelets. On the other hand, the electrical conductivity ($\sigma$) of the samples containing GnP-ZnO demonstrates superior values in comparison to the other composites containing GnP-$Fe_2O_3$ and GnP-$MnO_2$, correspondingly. The GnP-ZnO composite exhibits a maximum electrical conductivity of approximately 14.0 Scm$^{-1}$, which displays a positive correlation with temperature. This observation suggests the characteristic behavior of semiconductors. The composites containing GnP-$Fe_2O_3$ and GnP-$MnO_2$ exhibit a comparable pattern to that of GnP-ZnO, with the highest conductivity values being approximately 8.5 Scm$^{-1}$ and 5.5 Scm$^{-1}$, respectively. Even though the metallic oxides ($Fe_2O_3$, ZnO, and $MnO_2$) that were added are semiconductors, it is mostly the nanoparticles of these materials that give the samples their electrical conductivity along with the graphene nanoplatelets, and the latter portion makes the greatest contribution.

The measured Seebeck coefficient (S) values for each of the samples demonstrate a positive (+ve) indication, indicating that the composites that were synthesized possess characteristics of p-type semiconductors. This means that the hole as charge carriers are

crucial to their functioning in this regard. The maximum value of the Seebeck coefficient for the sample containing only the graphene nanoplatelets (GnP) is 30 $\mu VK^{-1}$ at 70 °C. Conversely, the composites containing GnP-Fe$_2$O$_3$, GnP-ZnO, and GnP-MnO$_2$ exhibit a maximum Seebeck coefficient of 105 $\mu VK^{-1}$ at 70 °C, approximately 140 $\mu VK^{-1}$ at 70 °C, and 100 $\mu VK^{-1}$ at 65 °C, respectively. The incorporation of nanostructured materials such as graphene nanoplatelets and metallic oxides in cement composites results in a higher Seebeck coefficient compared to samples without such additives, indicating the presence of an effective thermoelectric component of them. However, the greatest Seebeck coefficient improvement is attributed to GnP-ZnO samples, which is anticipated by a higher degree of contribution from ZnO [11] compared to graphene [8].

The determination of thermal conductivity ($\kappa$) is achieved through the correlation between thermal diffusivity and specific heat capacity. It is revealed that the thermal conductivity average values of the composite containing GnP, GnP-Fe$_2$O$_3$, GnP-ZnO, and GnP-MnO$_2$ are around 0.94, 0.65, 0.95, and 0.90 $Wm^{-1}K^{-1}$, respectively. The non-dimensional figure of merit (ZT) is derived using the formula $ZT = \frac{S^2 \sigma T}{\kappa}$. The cement composite containing GnP, GnP-Fe$_2$O$_3$, GnP-ZnO, and GnP-MnO$_2$ exhibits a maximum ZT value of $0.028 \times 10^{-2}$, $0.5 \times 10^{-2}$, $1.00 \times 10^{-2}$, and $0.2 \times 10^{-2}$, respectively. Table 1 presents a comprehensive list of the thermoelectric parameters that were obtained for the cement composites.

**Table 1.** TE properties of cement composites containing GnP and metallic oxides.

| Cement Composites Containing | Seebeck Coefficient, S ($\mu VK^{-1}$) | Electrical Conductivity, $\sigma$ (Scm$^{-1}$) | Thermal Conductivity, $\kappa$ (Wm$^{-1}K^{-1}$) | Figure of Merit, ZT, ($\times 10^{-2}$) | Semiconduc-Tor Type |
|---|---|---|---|---|---|
| GnP | 30 | 11.6 | 0.94 | 0.028 | p |
| GnP-Fe$_2$O$_3$ | 105 | 8.5 | 0.66 | 0.5 | P |
| GnP-ZnO | 140 | 14.0 | 0.95 | 1.0 | p |
| GnP-MnO$_2$ | 100 | 5.5 | 0.90 | 0.2 | p |

### 3.2. Analysis of Maximum Electrical Power Output

Power was calculated using the Seebeck values (S), electrical conductivity ($\sigma$), and the length (L) and area (A) of composites made of graphene nanoplatelets (GnP) and metallic oxide (Fe$_2$O$_3$/ZnO/MnO$_2$). Based on the thermoelectric parameters, Equation (9) can be used to figure out how much power the GnP-based cement composites with oxide inclusions can produce at their highest level, which is called P$_{max}$. As illustrated in Figure 3, the maximum power output (P$_{max}$) is observed to be 3.5 $\mu$W at approximately 70 °C when utilizing GnP-Fe$_2$O$_3$ integrations. Conversely, the employment of GnP-MnO$_2$ inclusions yields a P$_{max}$ of approximately 1 $\mu$W at 65 °C. Composites containing GnP-ZnO demonstrated a respective increase of 42% and 83% compared to composites containing GnP-Fe$_2$O$_3$ and GnP-MnO$_2$.

In order to assess the power output produced by the actual area and conduct a comparative analysis with other energy harvesting mechanisms, the value of P$_{max}$ is recalculated for the cement composites, taking into account the surface area of one square meter of the sample, as depicted in Figure 4. The data presented in this figure suggest that the incorporation of composites in pavements, roofs, or building surfaces can result in a power output of approximately 1.5 W per square meter, given a temperature differential of approximately 50 °C. This can be achieved through the use of solely p-type GnP-ZnO-added cement composites with a thickness of 10 mm. The estimated value surpasses the reported value of a thermoelectric generator (TEG) when utilizing a human body source, as evidenced by a recorded value of 0.285 $Wm^{-2}$ [19]. According to a study [20], the Seebeck coefficient at the intersection of a PN-junction created by p- and n-type cement composite was found to be higher than that of the individual components. Therefore, the integration of GnP-ZnO composite and n-type cement composites may result in a significant enhancement of the Seebeck coefficient, thereby enabling a potential increase in the total output power.

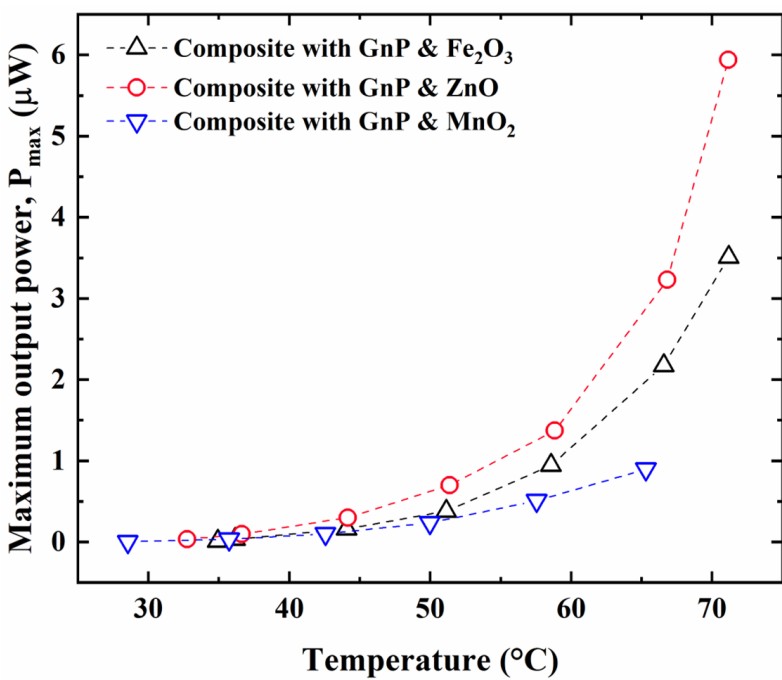

**Figure 3.** Maximum output power of cement composites containing graphene and metallic oxides.

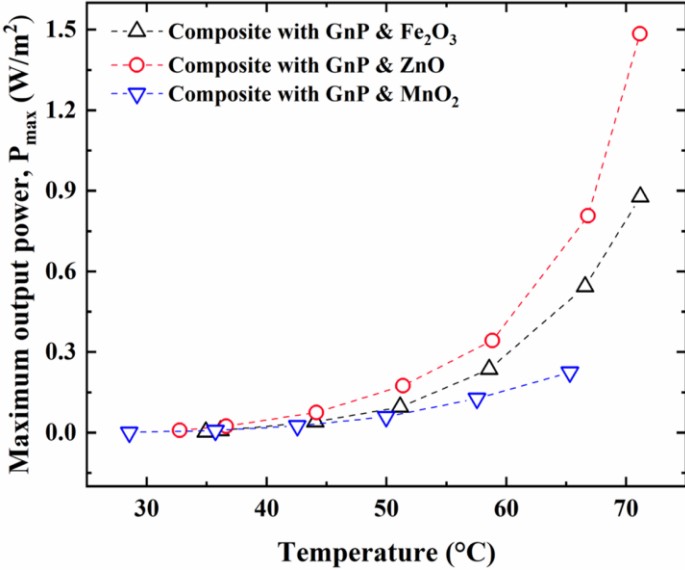

**Figure 4.** Normalized maximum output power/square meter w.r.to hot side temperature.

To derive an estimate of the output power generated by a thermoelectric generator (TEG) based on cement composites, it is necessary to utilize a combination of p-type and n-type composites. These composites must be arranged in a manner such that their junctions are electrically connected in series and thermally connected in parallel. This current study exclusively yielded a thermoelectric cement composite of p-type. Therefore, it is imperative to take into account the n-type. Wei et al. [5] have reported on a cement composite with n-type characteristics that exhibit a maximum Seebeck coefficient of approximately $-52\ \mu VK^{-1}$ at a temperature of around 70 °C, utilizing expanded graphite. This work is comparable in terms of loading concentration into the cement. The determination of the upper limit of electrical power for a structural thermoelectric generator has been conducted by utilizing the Seebeck coefficient of the n-type composite and our p-type composite (GnP and $Fe_2O_3$/ZnO/$MnO_2$), as outlined in reference [7]. According to Figure 5, the

fabrication of a cement composite apparatus utilizing an expanded graphite thermoelectric (TE) element of n-type and a graphene-ZnO TE element of p-type can yield a peak power output of 0.55 mW upon exposure to a thermal gradient of 50 °C.

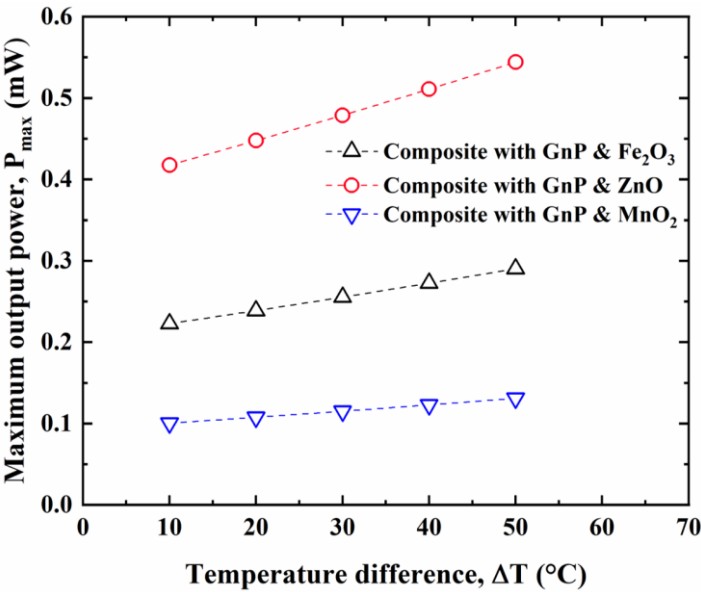

**Figure 5.** Maximum output power of a thermoelectric cement-based generator.

### 3.3. Analysis of Open-Circuit Voltage

The aforementioned peak power output has been computed utilizing the maximum power transfer theorem, which involves assuming that the resistance of the load on the outside is the same as the resistance inside the generator. Nevertheless, in pragmatic scenarios, a potential discrepancy may arise between internal and external resistance. Hence, the analysis of the open-circuit voltage of a thermoelectric device is of utmost importance. The highest Seebeck coefficient among the three metallic oxides ($Fe_2O_3$, ZnO, $MnO_2$) was observed in a p-type cement composite containing graphene and ZnO, with a value of $+141\ \mu VK^{-1}$. On the other hand, an n-type cement composite containing expanded graphite exhibited a Seebeck coefficient of $-52\ \mu VK^{-1}$ [5]. By comparing the values of the Seebeck coefficients of the p-type and n-type cement composites, one can calculate the maximum expected absolute output voltage at the open circuit. Specifically, a temperature gradient of 10 K would result in an output voltage of $|1.9\ \text{mV}|$ ($[|+141| + |-52|] \times 10 = 1.9\ \text{mV}$) [7]. However, to achieve a cement composite-based thermoelectric generator (TEG) with the capability to power functional devices, a minimum of 20 mV must be produced. The aforementioned value is derived from the requirement of voltage amplification through a voltage step-up converter in common use cases such as LED powering and energy storage in a capacitor. The LTC3108 manufactured by Analog Devices Inc., Wilmington, MA, USA, a DC-to-DC converter that operates without additional power requirement, is a commercially available ultralow voltage step-up converter, as reported in reference [21]. The converter has the capability to increase the output voltage from 2.2 V to 5 V, given an input of 20 mV. A Japanese research group [22] utilized a commercial converter to power an LED through a polymer-based TEG. The TEG comprises 300 thermoelements arranged in 10 parallel and 30 series, generating an output voltage of 40 mV. Based on the preceding information, it can be inferred that to achieve a voltage step-up conversion of at least 20 mV, approximately 22 thermoelements are necessary for our TEG when subjected to a temperature gradient of 10 K. Figure 6 depicts a potential configuration of thermoelements that alternate between p-type and n-type.

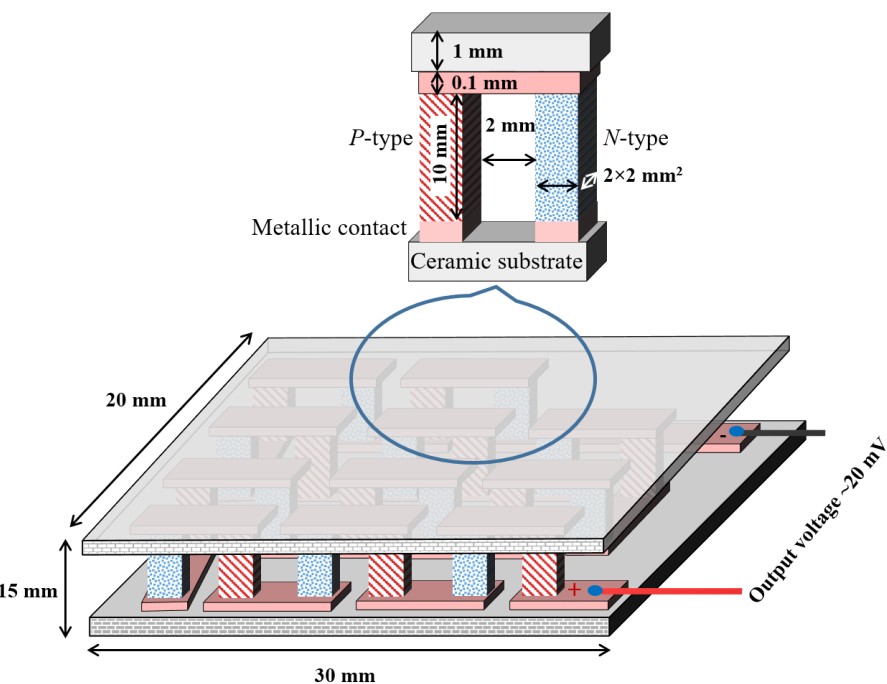

**Figure 6.** A cement composite-based TEG module's layout of its 22 thermoelements.

The utilization of cement composite thermoelectric generators (TEGs) that have a voltage output range of 20 to 50 mV, either with or without a step-up converter to drive the DC voltage produced in a capacitor, has the potential to provide power to low-consumption electronic devices that are integrated with buildings. These devices may include ultra-low-power microcontrollers and wireless sensor networks [23]. According to a study [24], the utilization of advanced devices such as smart home technologies and Internet of Things (IoT) applications can be facilitated if the voltage generated by the TE generator (TEG) exceeds 100 mV.

### 3.4. Analysis of Energy Conversion Efficiency

The thermoelectric efficiency of cement composites containing graphene nanoplatelets (GnP) and metallic oxides ($Fe_2O_3$/$ZnO$/$MnO_2$) has been determined by applying Equation (11). Figure 7 displays the efficiency plot of GnP-$Fe_2O_3$, GnP-$ZnO$, and GnP-$MnO_2$ cement composites while maintaining a constant cold side temperature of 25 °C and varying the hot side temperature. The data presented in this figure suggest that there is a positive correlation between the hot side temperature and the maximum efficiency, as the latter tends to increase as the former increases. The GnP-$ZnO$ composite exhibits the highest efficiency (0.04%), as expected, owing to its superior figure of merit in comparison to GnP-$Fe_2O_3$ and GnP-$MnO_2$. Nevertheless, the projected efficacy exhibits a notable disparity when juxtaposed with alternative energy harvesting mechanisms. The primary cause of the reduced efficiency is attributed to the limited temperature differential existing between the warmer and cooler regions. One potential solution for enhancing the energy conversion of cement composite TEG is to ensure an abundant heat supply. The figure presented in Figure 8 demonstrates that the maximum power conversion efficiency achievable at a temperature difference of 100 K is approximately 0.1%, with a corresponding ZT value of 0.01. The aforementioned level of efficacy is attained through the utilization of a polymer thermoelectric apparatus, which serves the purpose of energizing Light Emitting Diodes [22]. It is worth noting that the maximum thermoelectric efficiency ($\eta_{max}$) exhibits greater magnitude in the presence of high values of the thermoelectric figure of merit (ZT) and temperature difference ($\Delta T$). Achieving an improvement of at least one order of magnitude in the figure of merit for cement composite, as compared to the presently highest reported values [12], would lead to the attainment of an efficiency level of 1.0%. Thus, significant enhancement

in ZT is required for the practical implementation of thermoelectric devices based on cement composites.

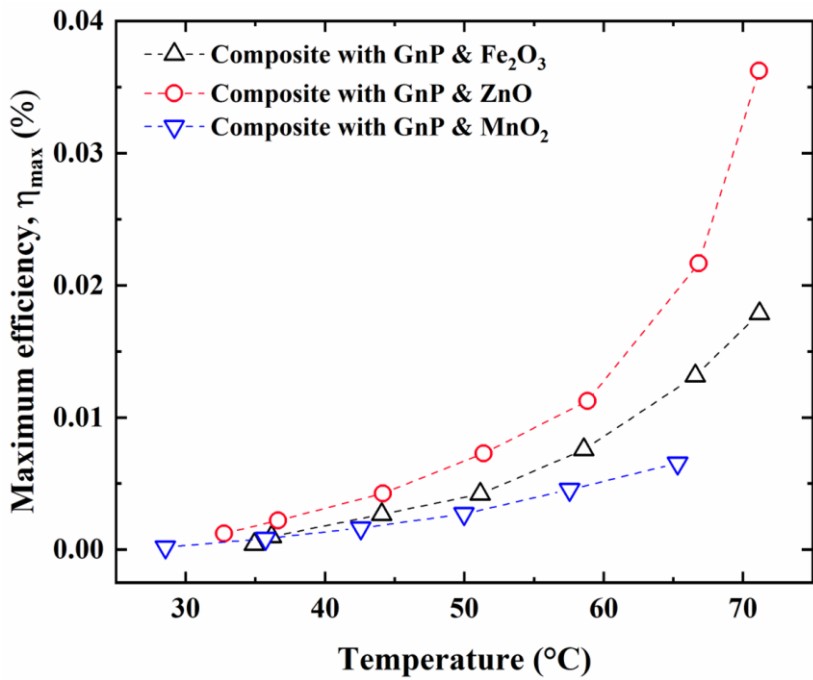

**Figure 7.** $\eta_{max}$ of GnP-oxides based cement composites for a constant (25 °C) cold side temperature.

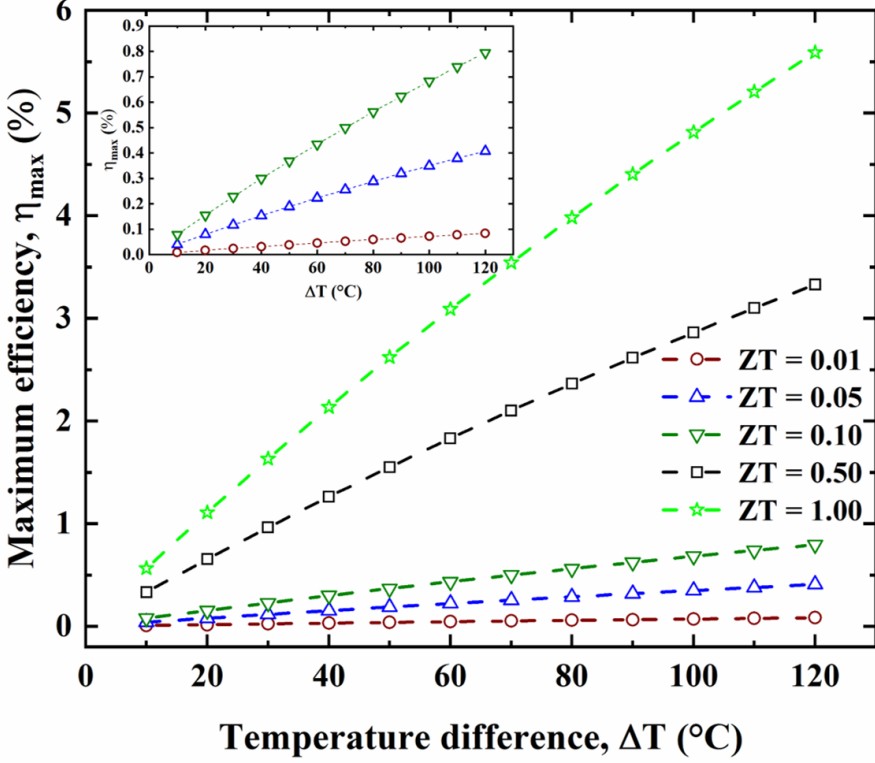

**Figure 8.** $\eta_{max}$ vs. figure of merit w.r.to temperature differences.

### 3.5. Possible Uses of Thermoelectric Cement Composite

The utilization of thermoelectric cooling technologies in buildings has garnered increased attention within the realm of energy conservation. In the realm of thermoelectric

cooling technologies, conventional thermoelectric materials, such as $Bi_2Te_3$, were utilized to construct thermoelectric devices that were subsequently affixed to the wall and window of a study [25]. However, if the concrete wall exhibits thermoelectric properties, utilizing its thermoelectric effect for energy generation and cooling would be a more efficient and practical approach. Thermoelectric cement composite devices exhibit promising potential for employment in the field of ambient energy harvesting. This technology can be used to harvest residual thermal energy from a wide variety of sources, including building envelopes, cities, highways, kitchens, and more. In a restaurant, for instance, the surface of a concrete wall next to the stove can get as hot as 60 or 70 degrees Celsius even though the air around it is just slightly warmer than room temperature or remains at room temperature. The thermal gradient presents a viable prospect for harnessing thermoelectric advancements to convert dissipated thermal energy into electrical power. Figure 9 illustrates a potential configuration of a wall composed of thermoelectric cement composite material for energy harvesting. For this purpose, the employment of thermoelectric cement composites in the form of thick rectangular slabs measuring $200 \times 100 \times 50$ mm$^3$ is considered.

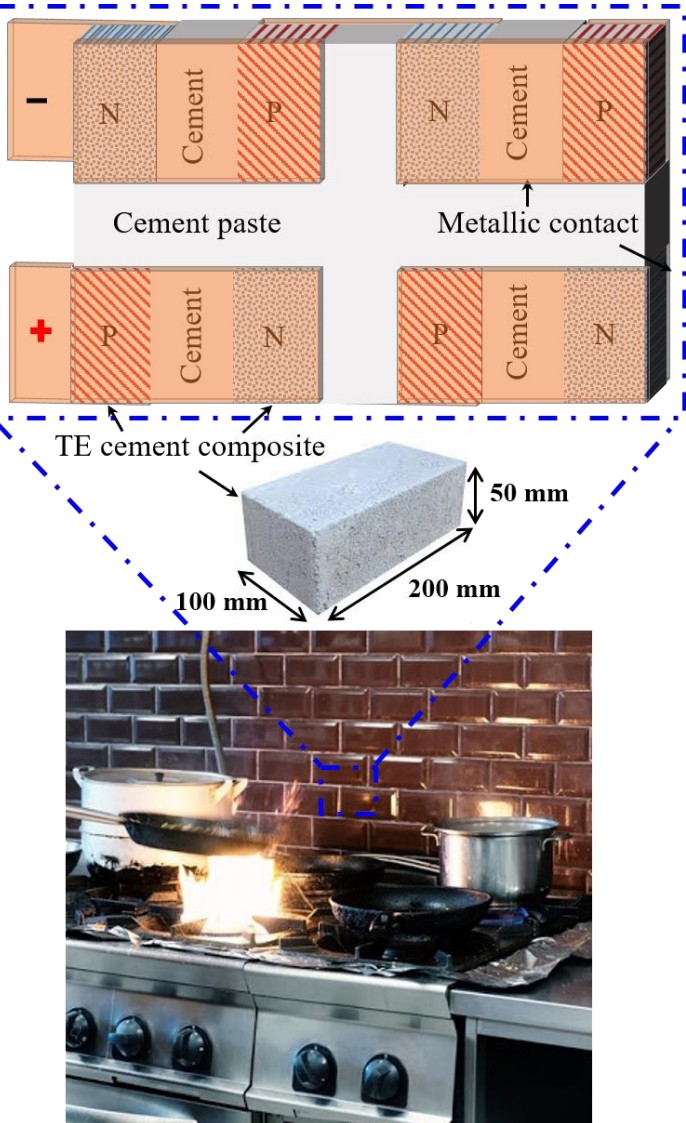

**Figure 9.** A potential application for energy harvesting by cement-based TEG.

## 4. Conclusions

Utilizing a structural cement composite as the basis for a thermoelectric generator presents promising prospects for harvesting ambient energy. This technology can capture and convert untapped thermal energy into usable electrical energy in various settings such as buildings, pavements, and industrial environments. A temperature differential of approximately 50 °C can result in the attainment of a power output of 1.5 W by a composite material consisting of graphene and ZnO added to cement, occupying an area of one square meter and having a height of 10 mm. It is possible to store the calculated energy within a capacitor and subsequently apply it within a wired or wireless communication network for various buildings. The incorporation of energy-saving and harvesting technology in buildings has the potential to facilitate the development of energy-efficient buildings and environmentally sustainable construction practices, leading to the creation of multifunctional smart buildings and civil structures.

**Author Contributions:** Conceptualization, S.G.; methodology, S.G.; software, B.B.S.; formal analysis, S.G., investigation, S.G.; resources, B.B.S.; writing—original draft preparation, S.G.; writing—review and editing, B.B.S.; visualization, S.G.; supervision, B.B.S. All authors have read and agreed to the published version of the manuscript.

**Funding:** This research received no external funding.

**Data Availability Statement:** Not applicable.

**Acknowledgments:** The authors are thankful to Michitaka Ohtaki, Interdisciplinary Graduate School of Engineering Sciences, Kyushu University, Japan, and Sivasankaran Harish, Department of Mechanical Engineering, The University of Tokyo, Japan for their valuable contributions.

**Conflicts of Interest:** The authors declare no conflict of interest. The funders had no role in the design of the study; in the collection, analyses, or interpretation of data; in the writing of the manuscript; or in the decision to publish the results.

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
