# Peer review of "Estimation of Energy Harvesting by Thermoelectric Cement Composites with Nanostructured Graphene and Metallic Oxides"

_jcs, doi:10.3390/jcs7050207_

Round 1
Reviewer 1 Report
This paper reported cement composite material consisting of graphene nanoplatelets and metallic oxides (Fe2O3, ZnO, MnO2), which can generate a maximum of 1.5 W 17 power per square meter for a temperature difference of about 50 °C using GnP-18 ZnO added cement composites. The research significance of this work is unclear and the underlying mechanism was not clearly established. I suggest this manuscript could be accepted after major revision. Some of the comments are summarized as follow:
1. The authors need to elaborate the novelty and significance of this study in comparison to previous reports.
2. TE materials with graphene nanoplatelets need to be included for comparison. Are the TE properties of the composites would be enhanced with the incorporation of metallic oxides? And why?
3. The mechanism behind the higher TE properties with the incorporation of ZnO compared with other metallic oxides was unclear.
4. How about the stability of the composites?
None.
Reviewer 2 Report
Ghosh and Saha report on the thermoelectric efficiency of modules made of cement composites, with graphene nanopowders and three different types of oxides. The study interesting and to a large extent well described.
There is an issue that is not clear: the Methods talks abouyt the preparation of the cement+mixture, and clearly states that it was further dried to remove any moisture.
What is then unclear is in what hydration state the composites were measured, as for their thermoelectric properties.
Cement, as a building material, is mixed with water, and its chemical and mecvhanical properties change drastically, dependent on the water-cement ratio, hydration conditions and times.
However, if the properties in this study were measured prior to hydration, then the relevance to building materials is doubtful. Also, in this case, how was a proper pellet prepared? Simply by cold-pressing as alluded to Fig1?
In addition, the mathematical development between Eq(1) and (11) is well known, although there is nothing wrong with reproducing it here.
Round 2
Reviewer 1 Report
This work can be accepted.
Reviewer 2 Report
the authors have adequately responded to my quueries and have edited the mansucript extensively.